# Droplet Distribution in a University Dental Clinic Setting: The Importance of High-Volume Evacuation

**DOI:** 10.3390/healthcare10091799

**Published:** 2022-09-19

**Authors:** Linda Gualtieri, Ronald Yong, Jessley Ah-Kion, Amanda L. A. Jamil, Asmae Bazaei, Jhanvi Kotecha, Sharron Long, Gloria Silcock, Catherine M. Miller

**Affiliations:** 1College of Medicine and Dentistry, James Cook University, Smithfield, QLD 4870, Australia; 2Australian Institute of Tropical Health and Medicine, James Cook University, Smithfield, QLD 4870, Australia

**Keywords:** droplet distribution, ultrasonic scaling, infection control, debridement high-volume evacuation, slow-volume evacuation crown preparation

## Abstract

The purpose of this study is to compare droplet distribution during a piezoelectric ultrasonic debridement procedure using either high-volume or slow-volume evacuation. Droplet distribution during a crown preparation with slow-volume evacuation is also examined. Fluorescein dye is added to the water reservoir and the procedures are performed by a single operator for 15 min on a dental manikin with artificial upper and lower teeth. Placement of filter paper squares (10 cm × 10 cm) in radiating lines away from the oral cavity of the dental manikin allows for visualization of droplet dispersion. Results show minimal difference in the spread of the droplets between the two evacuators during the debridement procedure; however, the slow-volume evacuator produces a higher concentration of droplets than the high-volume evacuator. An even higher concentration of droplets in the vicinity of the dental chair is observed during the crown preparation procedure. This study recommends the use of a high-volume evacuator where possible during professional debridement and crown preparation to reduce contamination around the dental chair from potentially pathogenic microorganisms.

## 1. Introduction

Routine dental procedures, including tooth preparation and ultrasonic scaling, produce copious amounts of droplets and splatter, identified as potential pathways for the spread of infection [1]. A debridement procedure routinely uses an ultrasonic instrument (scaler) to remove plaque and calculus [2]. The ultrasonic instrument uses water as a coolant, which is emitted through the oscillating tip, forming droplets and droplet nuclei. Droplets (splatter; >100 µm in size) settle in the vicinity of the source while droplet nuclei (<10 µm in size) can remain airborne for hours before settling [3]. These droplets and droplet nuclei mix with saliva and plaque, creating potentially infectious droplets that may become a major route of airborne transmission for infectious diseases [3]. A sampling of air during dental treatment has detected up to 418 colony-forming units (CFUs) per m^3^ [3]. Although a clear and direct link between dental treatment and the transmission of COVID-19 has not been identified, the SARS-CoV-2 (COVID-19) virus has been detected in saliva and can conceivably be spread via airborne droplets generated during clinical procedures [4,5,6]. This places individuals in close proximity to those receiving dental treatment, such as dentists, dental assistants, or nearby patients, at a high risk for infection [4,7,8]. This highlights the importance of proper infection control measures for each and every patient.

The design and layout of the dental clinic can play an important role in preventing infection [7]. This includes measures such as assigning designated areas of treatment (contaminated areas) and non-treatment (non-contaminated areas) areas as well as incorporating anti-microbial materials such as copper into frequently touched surfaces [7]. Infection control measures that aim to minimize droplet spread to patients, dental staff, and dental students also play an important role. The most conventional and effective way of reducing droplets and splatter is through high-volume evacuation (HVE). An HVE is a suction device mounted on an evacuation system that draws large volumes of air (up to 283 cubic centimeters of air) per minute [9]. In comparison, a slow-volume evacuation (SVE) is useful in dental procedures for removing pooling saliva and water during treatments. SVEs do not have the power necessary to limit the spread of droplets generated during procedures. The American Dental Association (ADA) recommends HVE use in all dental droplet-generating procedures for compliance with infection control protocols [10]. Due to the strong vacuum generated by the HVEs a dental assistant is necessary to stabilize the HVE in the oral cavity [10]. Due to the limited availability of dental assistants in the James Cook University Dental Clinic (JCU Dental), HVEs are seldom used during droplet-generating procedures, potentially compromising infection control.

Designed primarily as a teaching institute, the JCU Dental Clinic has an open floor plan to facilitate interactions between students and supervisors. As such, there is a capacity for droplets to extend beyond the treatment area immediately surrounding the patient. At the start of the COVID-19 pandemic, JCU Dental, along with many other healthcare facilities [11], introduced greater infection control measures to limit the risk of infection between patients, staff, and students. These control measures included removing stocks of gloves and masks from individual cubicles, switching to non-permeable isolation gowns for staff and students, and requiring patient consultations to be performed while wearing personal protective equipment (PPE). Further control measures included introducing pre-procedural diluted 3% hydrogen-peroxide mouth rinses and the compulsory use of HVE during droplet generating procedures.

The aim of this research project was to assess if infection control measures implemented around the containment of aerosols were sufficient. The specific objectives were the following: (1) to explore the extent of droplet spread during routine procedures within the JCU Dental clinic; (2) to examine if there was any significant decrease in droplet generation with HVE, in comparison to SVE in ultrasonic scaling or crown preparation procedures; (3) to determine places within the treatment areas where PPE could be placed without risk of contamination.

## 2. Materials and Methods

This study was conducted at JCU Dental, Cairns, Queensland, Australia. It is a teaching clinic with an open plan setting (OPS). Temperature and ventilation are controlled by air conditioning, providing a standardized environment. Cubicles are clustered into groups of eight, with four cubicles on each side of a central aisle. Each cubicle is 2.75 m wide and 4.27 m long and is separated by a 1.48 m high half-wall from the next cubicle. The cubicles are identical, each containing a dental chair, computer workstation, handwashing sink, and limited bench space. A dental manikin (generously provided by One Dental, Castle Hill, New South Wales, Australia) was placed in the chair and angled at approximately 30 degrees, known as the semi-supine position. A typodont (also generously provided by One Dental) was placed inside the mouth to replicate a patient’s natural dentition.

Squares of filter paper (10 cm × 10 cm) (Whatman 3M Filter Papers™ Grade 3 Sigma-Aldrich, Castle Hill, NSW, Australia) were placed around the manikin to collect droplets. Using the oral cavity of the manikin as the center six transects were placed at 12, 2, 4, 6, 8, and 10 o’clock positions. Tape was extended from the focal point for 1.8 m along the transect (Figure 1). The filter papers were placed 30 cm apart, along the tape, and secured in position. For the 6 o’clock position the filter papers were placed along the tape on the dental chair, where the patient would be situated. Filter papers were labeled according to their position along the tape and clock design (Figure 1). For example, a filter paper located in the 12 o’clock position at the third point along the tape was identified as “12.3”. Extra sheets of filter paper (297 × 420 mm) were placed along the bay’s walls to assess droplet spread to these areas. Additional filter papers were placed in various locations in the adjacent bay, which included the dental chair, the overhead light, and bench top.

The revolutions per minute (RPM) of the high-speed handpiece, ultrasonic instrument, and water flow rate were fixed and consistent throughout the study. The piezoelectric ultrasonic was set at 100 kHz and the water intensity set at maximum. Fluorescein dye (1 g/L; Sigma-Aldrich, Castle Hill, NSW, Australia) was added to the water in the dental unit to help visualize the droplets. The droplets fluoresced yellow when exposed to ultraviolet (UV) light. The same right-handed operator performed each procedure for 15 min.

### 2.1. Droplet Distribution during a Debridement

The spread of droplets generated from a piezoelectric ultrasonic hand instrument routinely used during a debridement procedure was investigated. The debridement was completed twice with SVE and twice with HVE. During the procedures with HVE, a second operator stood to the left of the dental manikin to hold the evacuator in place during the procedure.

Following each trial, the extent of droplet contamination was determined. A transparent grid containing 1 cm^2^ squares was placed over the filter paper and a UV torch used to visualize the fluorescent dye. The area of contamination was recorded by counting the number of contaminated 1 cm^2^ squares. Any squares present with at least one yellow area were considered contaminated. Data were recorded for each piece of filter paper and the area of contamination calculated. Surfaces outside the transects were inspected to determine droplet spread, if any, beyond the cubicle with the dental manikin.

### 2.2. Droplet Distribution during a Crown Preparation

A crown preparation procedure was also performed using a dental mirror, crown preparation burs, high-speed handpiece, and slow volume evacuator. Droplets were collected on filter paper in the same manner as described in Section 2.1. The distribution was compared with results obtained with the piezoelectric ultrasonic scaler and SVE to see if there was a greater distribution of droplets during this procedure.

## 3. Results

### 3.1. Droplet Distribution Is Reduced When HVE Is Used during a Debridement Procedure Compared with SVE

Figure 2 shows the extent of droplet distribution during the debridement with HVE (Figure 2a) compared with a debridement with SVE (Figure 2b), while Table 1 shows the area of droplet distribution on each filter paper square. The extent of the droplet distribution was similar between the two evacuation methods (Figure 2a,b), but the area of contamination with droplets found on the filter papers varied (Table 1). Splatter was found at 30 cm and 60 cm from the focal point for all the transects regardless of the evacuator used; however, the area of contamination on the filter paper was lower for the transects when HVE was used during the debridement rather than SVE (Table 1). Beyond 60 cm from the focal point, most of the splatter was found on the filter papers on the transects facing away from the operator (12 o’clock, 2 o’clock, and 4 o’clock). Limited distribution was observed along the patient (6 o’clock) or behind the operator (8 o’clock and 10 o’clock).

### 3.2. Area of Contamination with a Crown Preparation Was More Extensive Than a Debridement Using SVE

The extent of droplet distribution by the crown preparation procedure was similar to the distribution seen for the debridement with SVE (Figure 2b,c). However, the area of contamination was higher during the crown preparation procedure (Table 1). As with the debridement procedure, most of the splatter was found on the side facing away from the operator (12 o’clock, 2 o’clock, and 4 o’clock). Less extensive splatter was found along the patient and behind the operator, however, more splatter was observed in these areas than was seen during the debridement procedure. Large amounts of splatter were also observed on the dental manikin and typodont (Figure 3a) and the disposable gown of the operator (Figure 3b).

### 3.3. Droplet Distribution in Other Areas of the Cubicle

The filter paper used to cover the walls of the cubicle, including the area where the PPE was placed (Figure 1), was checked for any sign of droplet distribution, but no droplets were detected. Checking of filter paper laid in adjacent bays also showed a lack of droplets, indicating most droplets fell within the cubicle and relatively close to the dental chair.

## 4. Discussion

The COVID-19 pandemic has increased awareness and demand for aggressive infection control measures across the dental field. Droplets containing infectious microbes such as COVID-19 may spread disease [2]. The airborne transmission of COVID-19 emphasized the need to re-assess the infection control protocols in medical facilities and implement necessary modifications [12,13]. In response, there have been many studies investigating methods of reducing droplet spread [12,13,14]. Reducing aerosol contamination may reduce the risk of COVID-19 transmission and improve infection control measures [2,14].

Our study showed that most of the droplet spread was contained within a 1.5 m radius from the oral cavity for the two procedures tested; however, a small amount of splatter was found just outside the perimeter of the cubicle. This is consistent with findings in other studies conducted in both open-plan and closed-plan settings [3,15,16]. The droplets were not spread in a consistent circle around the chair. The sides of the cubicle away from the operator showed an increase in the number of droplets seen compared with the sides that were behind the operator. Droplets detected on the PPE the operator was wearing (Figure 3b) indicated the position of the operator was impeding the ability of the droplets to spread in that direction.

The surface area covered by droplets decreased as the distance from the mouth of the dental manikin increased, regardless of the evacuator or dental procedure. There was, however, a notable decrease in splatter generated when HVE was used compared with the use of SVE for debridement or crown preparation. The crown preparation produced larger droplets and increased saturation of the surrounding area compared with the debridement procedure. This highlights the effectiveness of using HVE to reduce aerosol contamination in the surrounding environment.

### Limitations

There are a few limitations to this study that could affect the interpretation of the results. The study took place in an open-plan dental clinic where temperature and humidity are centrally controlled. Ventilation in this type of building will be different from that in a clinic where it is possible to open windows and change temperature settings, and this could potentially affect how far the droplets could travel. Human subjects were not involved, so it presented circumstances that would not normally occur in a debridement procedure. The position of the manikin, degree of mouth opening, time allocated to debridement, and the setting of the piezoelectric ultrasonic did not necessarily represent a typical debridement procedure.

The position of the manikin was fixed, whereas under normal circumstances there would be more patient movement during the procedure. Another factor to consider is the degree of mouth opening. A reduced mouth opening would result in less splatter. Inversely, an increase in mouth opening would result in more splatters being propelled from the oral cavity. The manikin’s mouth opening was fairly limited and did not represent a patient’s mouth where the buccal mucosa could be easily stretched and widened for easier access and visibility. Moreover, the debridement experimental procedure was performed for 15 min. In practice, a dental cleaning can vary from 30 to 60 min depending on the amount of plaque, calculus, and periodontal status a patient presents with [5,17]. Moreover, the piezo-electric ultrasonic was not set at a typical standard setting of 40 kHz, as regulated by JCU Dental Clinic, because droplets were not visible at this setting. To accommodate this, the ultrasonic setting was adjusted to 100 kHz.

Despite these limitations, we feel the strength of our study is that it does give an insight into the spread of potentially infectious droplets during routine dental procedures and the protection from infection afforded by the use of HVE. The information generated in this study was used by the Management Committee of JCU Dental to formulate policies surrounding the use of HVE in high-risk procedures and the placement of PPE to avoid contamination.

Overall, our results suggest droplet production is still a concern with or without utilizing HVE. This study emphasizes the need for combining multiple infection control protocols to minimize aerosol contamination. Future studies may analyze droplet contamination with multiple bays operating simultaneously to replicate reality more closely and the complex interactions this may elicit [11]. This may include assessing ventilation positioning and adjusting airflow intensity. This study relied on the passive settling of aerosols on filter paper. Future studies can incorporate active sampling techniques; however, this could make mapping the distribution of contamination difficult [15]. Furthermore, studies involving the assessment of bacterial counts and microbiological analysis are needed since droplet production does not equate to contaminated droplet production [2].

## 5. Conclusions

Droplet splatter remains a prevalent issue in dental settings, as it increases the risk of disease transmission. With the existence of COVID-19, this issue has increased in significance. Our results suggest that the utilization of high-volume suction notably increases the effectiveness and efficiency of droplet distribution control. This emphasizes the necessity to utilize a dental assistant to assist in reducing the splatter of contaminated air particles.

## Figures and Tables

**Figure 1 healthcare-10-01799-f001:**
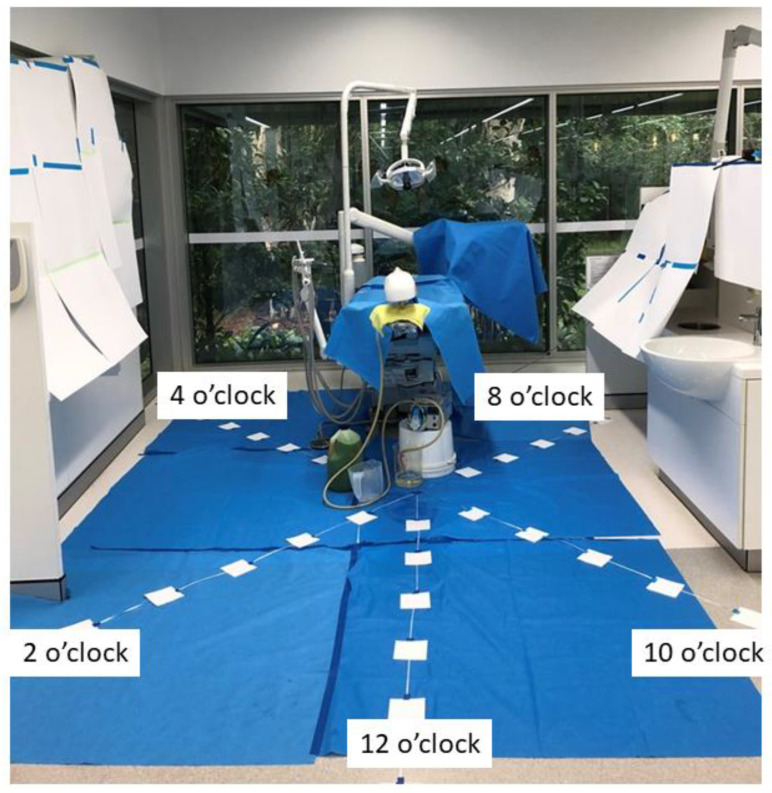
**Layout used for the experiments**. A dental manikin and typodont consisting of complete upper and lower plastic teeth were arranged in a dental chair as described in the “Materials and Methods”. Lines of tape were arranged radiating out from the head and filter paper laid out at 30 cm intervals. The transect coming away from the manikin was designated 12 o’clock and the others designated 2 o’clock, 4 o’clock, 6 o’clock, 8 o’clock, and 10 o’clock in a clockwise direction. Following a simulated debridement each filter paper was collected and analyzed for the presence of droplets as described in Materials and Methods.

**Figure 2 healthcare-10-01799-f002:**
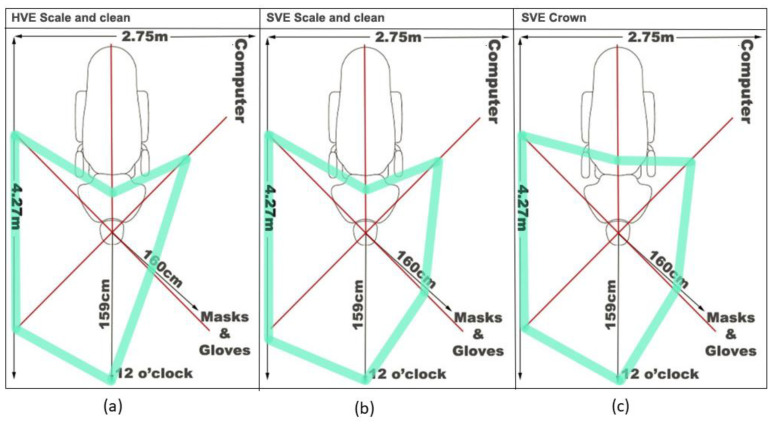
Distribution of droplets in the cubicle after the debridement and crown preparation procedures. A schematic representation of the extent of droplet distribution following (**a**) a debridement with HVE, (**b**) a debridement with SVE, and (**c**) a crown preparation. The typodont and dental manikin head are located at the point where the lines converge with the operator to the right between the 8 o’clock and 10 o’clock transects. The red lines show the transects where the filter paper squares were placed while the green line indicates the furthest point where dye was detected using UV light.

**Figure 3 healthcare-10-01799-f003:**
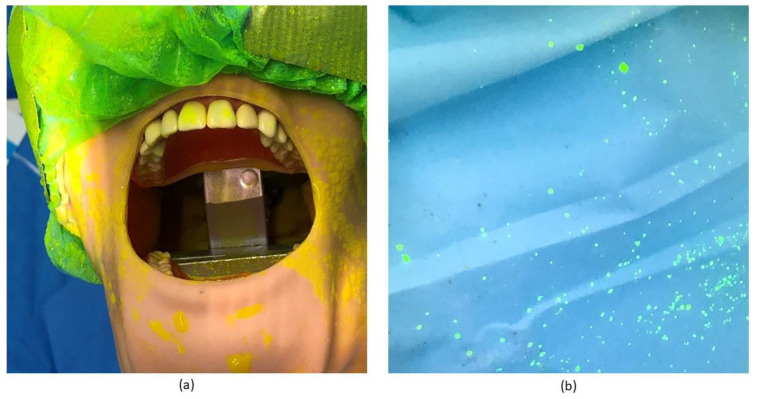
Extent of droplet splatter following crown preparation using SVE. Fluorescein dye was added to the water reservoir and a crown preparation simulated using SVE, as detailed in the Materials and Methods. Extensive droplet distribution was detected with UV light after the procedure was finished on (**a**) the dental manikin and typodont, and (**b**) on the disposable gown of the operator.

**Table 1 healthcare-10-01799-t001:** Area of filter paper covered with droplets following cleaning procedures (cm^2^).

Position of Filter Paper (O’Clock)	Area of Filter Paper Covered after Scale and Clean with HVE (cm^2^) *	Area of Filter Paper Covered after Scale and Clean with SVE (cm^2^) *	Area of Filter Paper Covered after Crown Preparation with SVE (cm^2^) *
2.1	83	98	100
2.2	14	98	100
2.3	1	87	100
2.4	5	69	100
2.5	3	10	100
2.6	0	5	100
4.1	100	100	98
4.2	86	99	98
4.3	4	84	92
4.4	2	66	94
4.5	10	48	80
4.6	2	14	21
6.1	3	94	100
6.2	3	5	98
6.3	0	0	89
6.4	0	0	0
6.5	0	0	0
6.6	0	0	0
8.1	4	19	84
8.2	1	4	24
8.3	0	3	22
8.4	17	1	4
8.5	0	0	0
8.6	0	0	0
10.1	42	100	24
10.2	1	61	14
10.3	0	9	5
10.4	0	0	0
10.5	0	0	0
10.6	0	0	0
12.1	53	90	83
12.2	49	47	85
12.3	33	33	77
12.4	15	16	85
12.5	18	3	92
12.6	10	3	0

* Average of results from two experiments.

## Data Availability

Not applicable.

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
