# Peer review of "Droplet Distribution in a University Dental Clinic Setting: The Importance of High-Volume Evacuation"

_healthcare, 2022, doi:10.3390/healthcare10091799_

Round 1
Reviewer 1 Report
The Paper is well written and straightforward. Although few questions need to be addressed prior recommending this article for publication:
1. Did the authors attempt testing by placing some king of a physical barrier around the month which may significantly impede the splatter on the floor. Perhaps a modifies dental might be worth investigating.
2. Would it be possible to modify the room humidity which might make the droplets heavy and restrict the range of their trajectories?
Author Response
- Did the authors attempt testing by placing some king of a physical barrier around the month which may significantly impede the splatter on the floor. Perhaps a modifies dental might be worth investigating.
Thank you for the suggestion. We originally chose to not use a rubber dam for crown prep procedures as they are not routinely used in our clinic but this is something we could potentially do in follow-up experiments. Our clinic has now installed a third hand ADEC high speed evacuation system so or follow up work would include the use of this system and we would also incorporate a physical barrier to impede splatter.
- Would it be possible to modify the room humidity which might make the droplets heavy and restrict the range of their trajectories?
This would be a good idea to test, unfortunately, the clinic is open plan with a centrally controlled temperature regulation system. The clinic is located in a tropical region of Australia and it is unlikely we would get permission to alter the room humidity in this way.
Reviewer 2 Report
I applaud the authors for conducting this innovative work, really impressive. Dental settings are always at high risk for aerosol and salivary droplet generation. Dentistry with an engineering approach has provided a unique niche here in the manuscript. However, there are some key references that would enrich the depth of the literature review. I have added the comments which will be necessary to address before further processing:
1. Introduction:
- There should be more references added to enrich the literature on the dental clinic setting and the risk of disease spreading. I recommend adding a few including these two which pioneered the thought that dental clinic architecture has a huge impact on disease spreading.
https://doi.org/10.1177/1937586720943992
https://doi.org/10.1177/19375867211060822
- Please separately write the section mentioning the aims and objectives of the study.
2. Materials and methods:
- Figure 1: Please make the text background white for the directions on the picture (4 o'clock and all others).
- Figure 2: Please enlarge the images, they are not clear.
3. Discussion:
- Please add a section on the limitations and strengths of the study.
Author Response
- Introduction:
- There should be more references added to enrich the literature on the dental clinic setting and the risk of disease spreading. I recommend adding a few including these two which pioneered the thought that dental clinic architecture has a huge impact on disease spreading.
https://doi.org/10.1177/1937586720943992
https://doi.org/10.1177/19375867211060822
Thank you for the suggested references. Information from these articles and a couple of others have been added to the introduction and have hopefully enriched it.
- Please separately write the section mentioning the aims and objectives of the study.
Lines 67-72 The aim and objectives have been placed in a separate paragraph and have been made more explicit.
- Materials and methods:
- Figure 1: Please make the text background white for the directions on the picture (4 o'clock and all others).
We have made this modification to the text in the figure.
- Figure 2: Please enlarge the images, they are not clear.
Figure 2 has been edited to enhance the quality of the image and make it clearer.
- Discussion:
- Please add a section on the limitations and strengths of the study.
Limitations and strengths are discussed on Lines 210-223
Round 2
Reviewer 1 Report
The authors have satisfactorily addressed all the comments of this reviewer. However, it would be advisable to include a "limitations" section in the paper to list all the limitations of the experiment and results. This is the interest of adding clarity for the readers who might contemplate reproducing these results as part of their own research.
Author Response
Reviewer comment: The authors have satisfactorily addressed all the comments of this reviewer. However, it would be advisable to include a "limitations" section in the paper to list all the limitations of the experiment and results. This is the interest of adding clarity for the readers who might contemplate reproducing these results as part of their own research.
Response: Thank you for your comment. We had included a paragraph that addressed limitations but have now created a section for this (Lines 229-254) to emphasise these points. We also added conditions within the clinic itself to the discussion of limitations so hopefully any interested readers would be able to reproduce our study.